# Nickel Complexes in C‒P Bond Formation

**DOI:** 10.3390/molecules26175283

**Published:** 2021-08-31

**Authors:** Almaz A. Zagidullin, Il’yas F. Sakhapov, Vasili A. Miluykov, Dmitry G. Yakhvarov

**Affiliations:** Arbuzov Institute of Organic and Physical Chemistry, FRC Kazan Scientific Center, Russian Academy of Sciences, Arbuzov Str. 8, 420088 Kazan, Russia; zagidullin@iopc.ru (A.A.Z.); sakhapovilyas@mail.ru (I.F.S.); miluykov@iopc.ru (V.A.M.)

**Keywords:** nickel complexes, organophosphorus compounds, catalysis, organic and inorganic phosphanes, phosphorylation, electrochemistry

## Abstract

This review is a comprehensive account of reactions with the participation of nickel complexes that result in the formation of carbon–phosphorus (C‒P) bonds. The catalytic and non-catalytic reactions with the participation of nickel complexes as the catalysts and the reagents are described. The various classes of starting compounds and the products formed are discussed individually. The several putative mechanisms of the nickel catalysed reactions are also included, thereby providing insights into both the synthetic and the mechanistic aspects of this phosphorus chemistry.

## 1. Introduction

Taking into account the use of organophosphorus compounds in organic synthesis and homogeneous catalysis, materials chemistry, agrochemical crop protection and pharmaceutical discovery, new methods for their synthesis hold particular significance [1,2,3,4,5]. Traditional methods to construct carbon–phosphorus (C−P) bonds—a key step in the synthesis of organophosphorus compounds—such as radical methods, anionic methods and the Arbuzov, Pudovik, Michaelis, Kabachnik–Fields, Abramov reactions are well known. These classical methodologies suffer from safety problems and limited scope, lack of selectivity and the use of protective groups that lead to additional stages of synthesis. Therefore, there has been growing interest in the development of transition-metal-catalyzed C−P bond construction as one of the most attractive method due to the safety, selectivity, high functional group tolerance and 100% atom economy provided by this approach [6,7,8,9].

Among the wide range of transition metals used for C−P bond construction, our attention has focused on nickel complexes. The development of organonickel chemistry has led to the discovery of several remarkable catalytic systems with excellent practical applications [10,11,12]. Actually, homogeneous Ni catalysis is currently experiencing a period of growing interest, resulting in numerous fascinating applications in synthetic organic and organophosphorus chemistry. Catalytic cycles with organonickel complexes as intermediates in many cases demonstrate high efficiency and include non-reactive organic and phosphorus substrates, but it is difficult to predict and control all reaction pathways [13]. From another point of view, cost-effectiveness is the undisputed driving force and great advantage behind the choice of nickel for catalytic applications, because comparing the cost of catalyst precursors for nickel and other noble metals shows a dramatic difference.

This review provides an overview of the nickel-catalyzed synthesis of phosphanes, phosphonium salts, phosphane oxides and phosphorus acid derivatives. This review focuses on the latest advances in applications of nickel complexes as an effective catalyst in C‒P bond formation, some aspects of the reaction mechanism and important advances in the asymmetric synthesis of organophosphorus compounds.

## 2. Synthesis of Tricoordinated Organophosphorus Compounds

### 2.1. Synthesis of Phosphanes Using Phosphane Chlorides

Phosphane chlorides are a useful class of *P*-coupling partners because the commonly used secondary phosphanes, their oxides or borane complexes are pyrophoric, require an additional reduction step or are incompatible with other functional groups. Cristau et al. reported initial efforts in the Ni-catalyzed cross-coupling of diphenylphosphane chloride with arylbromides [14]. Since the reaction yielded a mixture of phosphonium salts and triphenylphosphane oxides after workup, it has been evaluated that it is not synthetically useful.

An interesting version of cross-coupling with diphenylphosphane chloride catalyzed by NiCl_2_(dppe) (dppe—1,2-bis(diphenylphosphano)ethane) in the presence of stoichiometric amounts of metallic Zn was reported in [15] (Figure 1). In this reaction, zinc performs two functions: it reduces Ni(II) to Ni(0) and gives rise to zinc phosphide Ph_2_PZnCl, which reacts with ArylX. This methodology represents a convenient procedure for the preparation of different tertiary phosphanes, including the coupling of sterically hindered aryl halides or sulfonates that contain *ortho*-substituents as well as amide groups in the substrates. It should be noted that aryltrifluoromethanesulfonates provide higher yields (46–95%) than similar bromides.

Later, using this methodology, functionalized triarylphosphanes were obtained with good yields (55–86%) in a one-step reaction of an equimolar mixture of chlorodiphenylphosphane and aromatic bromides in NMP or DMF at 110 °C in the presence of zinc dust as a cheap reductant and NiBr_2_(bpy) (bpy—2,2′-bipyridine) as an efficient catalyst [16] (Figure 1). The main features of this versatile method are the simplicity of the reaction conditions and the compatibility with various functional groups.

Since then, (trimethylsilyl)diphenylphosphane was also employed as the phosphorus coupling partner. The optimized conditions for the Ni-catalyzed C–P cross-coupling reaction included NiCl_2_(PPh_3_)_2_ as the catalyst, ^t^BuOK as the base and Me_3_SiPPh_2_ as the phosphane reagent at 90 °C [17] (Figure 2).

Nickel- as well as palladium- and copper-catalyzed couplings of terminal alkynes with chlorophosphanes were developed later [18,19]. The Ni-catalyzed coupling of diarylchloro-, dialkylchloro-, aryldichloro- and trichlorophosphane (PCl_3_) with terminal acetylenes is a smooth transformation leading to a corresponding C-P coupling product in high yield (Figure 3). It should be noted that the couplings of aryldichlorophosphanes and trichlorophosphane (PCl_3_) are not selective, resulting in a mixture of mono-, di- and trisubstituted products.

From the mechanistic point of view, the authors claim the reaction to be a heteroanalogue of the *Sonogashira* cross-coupling [20]. The initial step of the catalytic cycle is the oxidative addition of the chlorophosphane to the Ni complex, forming a Ni-phosphido complex. Subsequent ligand exchange from chloride to acetylide gives a Ni-organic sigma complex, which then liberates phosphanoalkyne after reductive elimination (Figure 4).

### 2.2. Arylation and Vinylation of Secondary and Primary Phosphanes

The first example of secondary phosphanes used in transition metal-catalyzed cross-coupling reactions was demonstrated by Cristau et al., who explored the arylation of diphenylphosphane Ph_2_PH by aryl bromides in the presence of NiBr_2_ salt [14]. Upon the reaction of bromobenzene with diphenylphosphane in the presence of NiBr_2_, a mixture of triphenylphosphane (31%) and tetraphenylphosphonium bromide (19%) salt was obtained (Figure 5).

Later, Shulyupin et al. further broadened the synthetic applicability by employing (Ph_3_P)_2_NiCl_2_ or Ni(acac)_2_ (acac—acetylacetone) as efficient catalyst precursors in the phosphination of vinyl bromides and chlorides with diphenylphosphane (Figure 6) [21]. The procedure uses a combination of up to 1 mol% of Ni complexes, triethylamine and DMF as a solvent, leading to products with 75–96% yields. The double bond geometry of the vinyl halides was retained under the reaction conditions.

Functionalized vinylphosphanes containing alkoxy- or amino- groups were synthesized by the reaction of diphenylphosphane or its trimethylsilyl derivative with the corresponding alkenyl bromides or chlorides under catalysis by nickel complexes. Ni complexes turned out to be more efficient than Pd complexes in reactions with less active alkenyl halides, such as 2-bromobutene and 1-bromovinylsilane [22]. Likewise, two diphenylphosphano groups were introduced into 1 and 4 positions of 1,4-diiodo-butadienes with 90–95% yields [23] (Figure 7).

A striking example of a Ni-catalyzed cross-coupling was developed by Cai et al. for the synthesis of BINAP [24], one of the most efficient and successful chiral ligands, which was synthesized for the first time by the Noyori group in the early 1980s. These authors anticipated the use of a Ni catalyst, instead of a Pd catalyst, to be more promising since the former binds to BINAP weaker than any other transition metal of the second or third row. Thus, catalyst poisoning was prevented. After the initial optimization of the reaction conditions, BINAP was synthesized without racemization. The best yield of 77% was obtained by NiCl_2_(dppe) cross-coupling of aryltriflate and diphenylphosphane in DMF in the presence of DABCO as a base [25], while other systems did not promote the reaction or led to side or oxidation reactions (Figure 8). Fortunately, Cai’s method is not restricted to diarylphosphanes, since dialkylphosphanes were also used in this reaction [26]. In addition, Wills and co-workers reported that, in some cases, the addition of zinc dust improves the yield [27].

Several research groups have adopted the Ni-catalyzed cross-coupling protocol developed by Cai et al. for the synthesis of a wide variety of chiral phosphanes: axially chiral Quinazolinap ligand [28,29], *P*-stereogenic BINAP [30] and other binaphthyl-based phosphane and phosphite ligands [31]. Figure 9 shows selected examples, such as steroidal [32] and pyrazinylnaphthyl derivatives of BINAP [33], PINAPs [34,35], or fluoroalkyl-tagged binaphthyls [36,37].

Recently, Zhao et al. disclosed a method for the cross-coupling of various aryl bromides with diphenylphosphane in the absence of external reductants and supporting ligands (Figure 10) [38]. The reaction gave a mixture of phosphanes and phosphane oxides with 64–99% yields. Several functional groups (ester, ether, ketone and cyano groups) remained intact under the conditions.

In addition to the above-mentioned methods, an electrochemically promoted nickel catalysed processes were also developed [39,40]. Organonickel sigma-complexes have been found as efficient key intermediates in various Ni-catalyzed C‒C coupling reactions [41], electrocatalytic processes and C‒P bond formation with participation of unfunctionalized organic arylhalides with elemental (white) phosphorus P_4_ [42], chlorophosphanes or various primary and secondary phosphanes (Figure 11) [43,44].

### 2.3. Hydrophosphination Reactions of Alkenes and Alkynes

Hydrophosphination reactions involve the addition of P–H to an unsaturated C–C bond and have gained great interest as an alternative to the classical phosphane syntheses involving a substitution that is incompatible with certain functional groups. In this reaction, phosphanes, silylphosphanes or phosphane–borane complexes are used as phosphinating agents to react with inactivated or activated alkenes, dienes and alkynes. Moreover, the addition of P–H to an unsaturated C–C bond is more efficient than substitution reactions when considering atom efficiency that makes it greener and more economical.

In recent years, great progress has been made in metal-complex-catalyzed hydrophosphination reactions [9,45]. It should be noted that reactions of P‒H and P‒E compounds with alkynes in the presence of transition metal complexes occur preferentially as *syn*-addition. It was shown that hydrophosphination reactions catalyzed by Ni-based complexes proceed more efficiently and allow inactivated alkenes to be employed [46,47,48,49]. Shulupin et al. first reported hydrophosphination with high yields up to 99% of weakly activated aryl olefins and their heterocyclic analogues in the presence of Ni(0) phosphite complexes (Figure 12) [50].

The reaction is regioselective: the only product is the corresponding *β*-phosphorylated adduct. The fact that no α-adduct is formed in the addition of Ph_2_PH to styrenes and vinylpyridines allows the formation of π-allyl intermediates to be excluded. A probable catalytic cycle includes the oxidative addition of phosphane to Ni(0) with the formation of a hydride phosphide complex, alkene insertion into the Ni‒H bond and subsequent reductive elimination (Figure 13). Later, Ganushevich et al. characterized, for the first time, the product of an oxidative addition of primary phosphane to a nickel(0) complex. The terminal phosphanido hydride nickel complex, [NiH{P(Dmp)(H)}(dtbpe)], where Dmp—2,6-dimesitylphenyl and dtbpe—1,2-bis(di-tert-butylphosphino)ethane, has been formed in this process [51,52,53]. 

Chiral metal complexes have been used to promote and control the asymmetric P–H addition reaction. A chiral pincer bisphosphane Ni complex was used in the first highly enantioselective catalytic synthesis of *P*-stereogenic secondary phosphane–boranes by the asymmetric hydrophosphination of electron-deficient alkenes with phenylphosphane. Various *P*-stereogenic secondary phosphane–boranes were obtained in 57–92% yields with up to 99% *ee* [54]. The Togni group developed the asymmetric hydrophosphination of vinyl nitriles catalyzed by a dicationic Ni-based complex yielding the desired phosphane product with 32–94% *ee* [55,56] (Figure 14).

The addition of P–H to a triple bond is a highly desirable method when considering the principles of atom economy. The first example of the hydrophosphination of terminal and internal alkynes, catalyzed by Pd and Ni complexes, was reported by Kazankova et al. (Figure 15) [57]. The regioselectivity was strongly dependent on the catalytic precursor and alkyne nature. In the presence of Pd(0) and Ni(0) complexes, the β-adduct was formed as the major product. By contrast, Pd(II) and Ni(II) complexes mainly gave rise to the α-adduct (α:β = 95:5) [58]. The different selectivity in the reactions catalyzed by Pd(0)/Ni(0) complexes and Pd(Ni)X_2_ is explained by the formation of catalytic amounts of HX (HOAc or HBr) in situ, which initiate the second catalytic cycle. The Ni-based catalyst was more effective than the Pd-based catalyst, and the reaction proceeded at lower temperatures. The relative reactivity of the metal complexes in the hydrophosphination of alkynes was studied theoretically by Ananikov et al., and it decreased in the order of Ni > Pd > Rh > Pt. [59]. The estimated relative reactivity order of the studied metals implies that nickel can not only be a cost-economic replacement of Pd, but also superior in terms of catalytic efficiency. In the reaction of diphenylphosphane with *tert*-butylacetylene, the corresponding β-adduct with 95% yield is formed as the only product for steric reasons. The addition of diphenylphosphane to other alkylacetylenes is characterized by lower selectivity, with both regio- and stereoselectivity strongly dependent on the reaction conditions.

Later, an efficient NCC pincer Ni(II)-catalyzed hydrophosphination of nitroalkenes with HPPH_2_ was developed. After the optimization of reaction conditions, (hetero)aromatic and aliphatic nitroalkenes were well tolerated, irrespective of electronic effects, to provide the products in up to 99% yield [60]. In addition to P(III) phosphanes, Montchamp and co-workers produced vinyl-*H*-phosphinates from alkynes and alkyl phosphinates using only 2–3 mol% NiCl_2_ [61]. Ananikov et al. have shown the Markovnikov-selective phosphorylation of internal and terminal alkynes using a range of phosphites in the presence of catalytic amounts of Ni(acac)_2_ and 1,2-bis(diphenylphosphano)ethane (dppe) [62]. Han et al. reported that nickel catalysts are more reactive than noble metal catalysts in the catalytic additions of a variety of P(O)−H bonds to alkynes (propargyl alcohols, 1-octyne), regioselectively affording both the Markovnikov and the *anti*-Markovnikov products in high yields (72–96%) [63]. A related five-coordinated hydrido nickel complex is successfully isolated in the catalysis, which can react readily with an alkyne to give the addition products (Figure 16) [64].

## 3. Synthesis of Tetracoordinated Organophosphorus Compounds

### 3.1. Synthesis of Phosphonium Salts in the Presence of Ni Salts

The reaction of aryl halides with PPh_3_ in the presence of Ni(II) salts is one of the oldest transition metal-catalyzed reactions known to form a C‒P bond (Figure 17). Iodobenzene worked best in the reaction (yield up to 90%), while the corresponding bromides and chlorides were less reactive and gave lower yields (60–90%). Electron-donating alkyl, amino or alkoxy groups facilitate the C‒P cross-coupling, while electron-withdrawing groups act as strong inhibitors [65]. The reaction is quite common for *para*- and *meta*-substituted aryl halides. Although *ortho*-substituted aryl halides are usually transferred due to the stabilization of the intermediately formed C_sp2_–Ni bond, [65] Allen et al. used the directing effect of an imine or a diazo nitrogen atom to perform a chemoselective cross-coupling reaction in the *ortho*-position under mild conditions [66].

Thiophene [67], furan [66] and pyrrole [68] halides also served as the substrates or as substituents at the phosphorus atom. In the case of trialkylphosphanes, high temperatures might be problematic for these sensitive substances, so further cross-coupling reactions were successfully performed in refluxing ethanol using (Ph_3_P)_3_Ni [65,69].

Later, the Charettea group reported a general and efficient Ni-catalyzed cross-coupling reaction between aryl halides (iodides, bromides, chlorides) or triflates and PPh_3_ generating tetraarylphosphonium salts in good to high yields (63–99%) (Figure 18). This Ni-catalyzed C‒P coupling is conducted in ethylene glycol using a readily available, cheap NiBr_2_ precatalyst and tolerates different functional groups such as alcohols, amides, ketones, aldehydes, phenols and amines [70].

Additionally, nickel-catalyzed C–P coupling polymerization of commercial aryl dihalides and diphenylphosphane was used for the convenient preparation of tetraarylphosphonium polyelectrolytes. A NiBr_2_-based catalyst was effective in C–P coupling reactions to yield tetraarylphosphonium polymers with degrees of polymerization up to about 30 [71].

### 3.2. Synthesis of Phosphane Oxides by C‒P Cross-Coupling

Transition metal-catalyzed C–P bond formation has been well explored [72]. In the last decade, there have been reports of the use of nickel-based catalysts for the synthesis of phosphane oxides. Yang and co-workers reported the synthesis of diphenylphosphoryl compounds through Ni-catalyzed cross-coupling of diphenylphosphane oxide Ph_2_P(O)H with heteroaryl chlorides (Figure 19) [73]. The reactions of various aryl halides with diphenylphosphane oxide were also carried out using the Ni(Zn) catalyst together with *N*-ligands in water, leading to the formation of diphenylarylphosphane oxides with 75–97% yields [74].

The Zhao group reported the Ni-catalyzed cross-coupling of functionalized arylboronic acids with *H*-phosphites, *H*-phosphane oxides and *H*-phosphinate esters to give various organophosphorus compounds with good to excellent yields (50–99%) (Figure 20) [75]. This strategy provided a generalized and substantial tool for the synthesis of triarylphosphane oxides and was the first example of a Ni-catalyzed C–P cross-coupling reaction utilizing >P(O)H substrates and arylboronic acids.

Later, Liu et al. demonstrated the reaction of 1,1-dibromo-1-alkenes with diphenylphosphane oxide using NiBr_2_(bpy) and magnesium in the presence of potassium phosphate at moderate temperatures (Figure 21) [76]. Mechanistic studies show that the reaction involves the «Hirao type» reduction to give alkenyl bromides, which undergo a reaction with diphenylphosphane oxide in the presence of Ni(0), which can be obtained from NiBr_2_ by reduction using magnesium.

The Xiao group reported the coupling of aryl iodides with diphenyl phosphane oxide at room temperature on the base of nickel and photoredox-based catalytic systems [77]. Photoredox catalysis using Ru(bpy)_3_Cl_2_(H_2_O)_6_ with 3W blue LED gives the *P*-centered radical, which reacts with the nickel(II)–aryl complex. The reaction takes place at room temperature and tolerates phenol, amide and ether functional groups.

Recently, the decarbonylative coupling of aryl esters with diphenylphosphane oxides in the presence of Ni(OAc)_2_ was demonstrated by Yamaguchi and co-workers (Figure 22) [78]. The key success of the transformation is the use of the 3,4-bis(dicyclohexylphosphano)thiophene (dcypt) ligand, and the yield was modest to good (48–82%).

A Ni-catalyzed asymmetric allylation of secondary phosphane oxides for the synthesis of tertiary phosphane oxides was realized in the Zhang group with high enantioselectivity (87–95% *ee*) (Figure 23) [79]. The protocol represents the first example of synthesizing *P*-stereogenic phosphane oxides by an allylation reaction. The finding of this research expands the applications of Ni-based catalysts and secondary phosphane oxides in the synthesis of *P*-stereogenic phosphanes.

Recently, a visible-light-induced Ni-based catalyst C–P coupling reaction of diarylphosphane oxides with aryl halides has been developed by the Zhu group. The Ni(I) species and chlorine atom radical were generated via the ligand to the metal charge transfer process of NiCl_2_(PPh_3_)_2_, which allows the formation of Ni(IV)–P species, giving various tertiary phosphane oxides under photocatalyst-free conditions at room temperature in good yields (40–75%) [80].

Aside from homogeneous systems, effort was made on the development of heterogeneous systems based on Ni/CeO_2_ or Ni/Al_2_O_3_ nanocatalysts for the coupling of aryl iodides or bromides with diphenyl phosphane oxides in the presence of K_2_CO_3_ [81]. The reaction is scalable and the nanocatalyst can be recycled without loss of activity.

### 3.3. Synthetic Routes of Phosphinates

Historically, one of the first methods of non-catalytic C‒P bond formation was reactions of alkyl(phenyl)phosphonous esters with aryl halides. One of the first catalytic versions of this reaction was performed with nickel complexes and aryl(alkyl)phosphonous esters with aryl bromides to give various phenyl-arylphosphinates with good to excellent yields (52–82%) (Figure 24) [82,83].

The carbon–phosphorus cross-coupling of aryl tosylates or mesylates has been accomplished with high yields up to 92%, with ethyl phenylphosphinate using NiCl_2_(dppf)/dppf (dppf—1,1′-bis(diphenylphosphino)ferrocene) at 100 °C in the presence of diethylisopropylamine (DIPEA) and zinc (Figure 25) [84]. The reaction can also be extended to the coupling of diarylphosphane oxides and diethylphosphonate with aryl sulfonates.

Later, Gao and co-workers achieved the coupling of phenylboronic acid with ethyl phenylphosphinate employing a NiBr_2_/pyridine system in the presence of K_2_CO_3_ (73% yield) (Figure 26) [75]. The reaction can be extended to the coupling of a series of phosphites and phosphane oxides.

### 3.4. Synthesis of Phosphonates by C‒P Cross-Coupling

#### 3.4.1. Ni-Catalyzed Phosphonylation with Phosphites

The reaction of trialkyl phosphites with alkyl halides (Arbuzov reaction) leads to the formation of phosphonates RP(O)(OR’)_2_. The Ni-catalyzed procedure using trialkyl phosphites or dialkyl arylphosphonites was described fifty years ago by Tavs et al. and allowed the involvement of aryl and alkenyl halides in this reaction [85]. This work is the first example of a transition metal-catalyzed cross-coupling reaction of C–P bond formation. The reaction requires harsh conditions (150–200 °C), but the yields of aryl- and alkenylphosphonates are high. The reaction proceeded smoothly using aryl iodides or bromides and phosphite and phosphonite ethyl esters (Figure 27). Exceptions are reactions with *ortho*-substituted aryl halides, where the yield decreases to 15–40%. Even alkenyl chlorides, such as α- and β-chlorostyrenes and vinyl chloride, give rise to alkenylphosphonates in high yields under NiCl_2_ catalysis [86]. Phosphinites also reacted with good yields [87].

The catalytic arylation of tris(trimethysilyl)phosphite occurs with higher rates (Figure 28) [88]. The treatment of the obtained arylphosphonates with methanol at room temperature gives the corresponding phosphonic acids in quantitative yield.

The reactions of triethylphosphite with akenylhalides having an alkoxy or diethylamino group in position 1 occurred under milder conditions (Figure 29) [89]. Similar reactions with 2-halovinyl ethers or 2-haloenamines proceeded at a higher temperature, but the yields were also quite high [90].

Another example is the conversion of (*E,E*)-1,3-diiodobutadiene into bis-1,4-(diethoxyphosphanoyl)-1,3-butadiene in high yields [23]. In the above reactions, triethyl phosphite acts as both a phosphorylating and a reducing agent. Balthazor showed that the reaction involves the formation of a Ni(0) phosphite complex in the presence of a slight excess of the trialkyl phosphite [91]. This phosphite Ni(0) complex undergoes fast oxidative addition with the aryl(vinyl) iodide, followed by slow decomposition to give the *quasi*-phosphonium salt, which is finally transformed into arylphosphonate via classical Arbuzov rearrangement (Figure 30).

Heinicke et al. described the reaction of triethylphosphite with 2-haloanilides in the presence of NiX_2_ complexes. The resulting *o*-acylaminophenylphosphonates are useful intermediates in the synthesis of 1*H*-1,3-benzazaphospholes (Figure 31) [92].

An efficient method has been developed for the Ni-catalyzed phosphonylation of aryl triflates with triethylphosphite, in which KBr as an additive promotes the S_N_2 catalytic step (Figure 32). This is the first example of nickel-catalyzed Arbuzov-type reaction of aryl triflates. Most of the substrates showed good reactivity with the use of these catalytic systems and good to high yields (46–95%) [93].

#### 3.4.2. Phosphonylation by Hirao Reaction

The Hirao reaction is an alternative synthetic method for obtaining a wide range of phosphonates, which proceeds under milder reaction conditions and with higher yields. It can be considered as a classic example of C‒P cross-coupling with the formation of a C_sp2_–P bond.

In the 1980s, Hirao et al. reported the first Pd-catalyzed cross-coupling reaction of dialkyl phosphites with aryl and vinyl bromides, resulting in dialkyl arylphosphonates and dialkyl vinylphosphonates, respectively [94]. Later, reactions of allyl acetates and allyl carbonates with dialkyl phosphites were investigated under Ni catalysis in the presence of bis(trimethylsilyl)acetamide (BSA) as a base [95,96]. A direct comparison of nickel- and palladium-catalyzed cross-coupling for vinyl halides was reported by Beletskaya and co-workers [90]. Aryl and vinyl iodides as well as bromides reacted smoothly under these reaction conditions, while the corresponding chlorides were unreactive.

Nowadays, nickel catalysis is often applied in Hirao C–P cross-coupling reactions, and a range of organic and organometallic compounds including organohalides, alcohol or phenol derivatives [97,98,99,100], aryl, benzyl or allyl ammonium salts [101], sulfides [102] and aryl nitriles [103,104] have been employed as the carbon coupling partners. Reductive procedures involve Ni(II) salts together with Zn/Mg as the reductant or without reductive agents or Ni(0)(cod)_2_ as the catalyst precursor [75].

Han and co-workers extended the substrate scope to aryl bromides and chlorides with dimethylphosphite and diphenylphosphane oxide using NiCl_2_(dppp) (where dppp—1,3-bis(diphenylphosphano)propane) in the presence of potassium phosphite K_3_PO_4_, leading to phosphonates with 50–96% yields (Figure 33) [38].

Challenging phenol derivatives could also be involved in C‒P coupling reactions after converting the hydroxyl function to a better leaving group by reaction with bromotripyrrolidinophosphonium hexafluorophosphate (PyBroP) (Figure 34) [105]. The method allows the C–P cross-coupling to be carried out in a one-pot procedure without the isolation of an activated phenol intermediate.

The substrate scope is further expanded for the coupling of aryl mesylates with dimethylphosphite and diphenylphosphane oxide utilizing NiCl_2_(dppf)/dppf at 100 °C in the presence of diisopropylamine and zinc [84]. Later, the decarboxylative coupling of alkenyl acids with *H*-phosphonates was shown to obtain (*E*)-1-alkenylphosphonates (Figure 35) [106]. The reaction utilizes NiCl_2_(dppf) with Ag_2_O at 100 °C under a nitrogen atmosphere. The substrate scope can be extended to the coupling of alkynyl acids to produce alkynyl phosphonates in moderate yields up to 92%.

A catalytic deamidative phosphorylation of a wide range of amides using a Ni catalyst giving aryl phosphonates in good to excellent yields was reported (Figure 36) [107]. This method tolerates a wide range of functional groups. The reaction constitutes the first example of the transition metal-catalyzed generation of a C‒P bond from amides.

Keglevich and co-workers found that NiCl_2_ may also be a suitable catalyst in the microwave-assisted C–P coupling of bromobenzene and different >P(O)H species [108]. The experiments were carried out at 150 °C under MW irradiation, applying K_2_CO_3_ in the absence of any solvent with 68–92% (Figure 37). The NiCl_2_-catalyzed phosphonylation of substituted bromoarenes led to similar results as in the presence of Pd(OAc)_2_, but the scope of the aryl bromides was somewhat limited.

Taking into account the reaction conditions, costs and safety concerns, it can be concluded that the application of Pd(OAc)_2_ is favorable, but the use of NiCl_2_ can also be a good alternative. Moreover, the C–P coupling reactions which apply Ni(II) salts in the absence of reductants have been investigated earlier, including theoretical calculations [109,110]. These latest developments of Hirao coupling mean a big step forward to “*P*-ligand-free” catalytic reactions, since there is no need for sensitive and expensive *P*-ligands.

Recently, Budnikova and co-workers have demonstrated the possibility of the electrochemical phosphorylation of aromatic compounds (benzene and coumarins [111], pyridines [112], azoles [113]) with dialkyl phosphites (Figure 38). This novel approach is based on the oxidation of a mixture of the aromatic compound and diethyl phosphite (1:1) under mild electrochemical conditions (room temperature, atmospheric pressure) in the presence of bimetallic catalytic systems: 1% of Mn(II)(bpy)/Ni(BF_4_)(bpy). This method allows one to obtain diethyl arylphosphonates in good yields (up to 70%) and 100% conversion of the phosphite [114].

## 4. Summary and Outlook

C‒P cross-coupling reactions have made significant progress in recent years. Although Pd-catalyzed reactions dominate, Ni-based catalytic systems have been considerably explored. This article may give a good overview on the present state of the art of the Ni-catalyzed synthesis of racemic and scalemic phosphanes, phosphonium salts, phosphane oxides and phosphorus acid derivatives. Additionally, some green chemical approaches, such as MW activation, solvent- and reducing agent-free and electrochemical methods, have been outlined. The renaissance in nickel catalysis has brought new life to well-known nickel salts NiX_2_ (X=Cl, Br, OAc, acac, etc.), which have been used as catalyst precursors. Solubility in organic solvents and the easy transformation of Ni(acac)_2_ to the catalyst active form ensure important preferences for practical applications.

The increasing price of Pd, Pt and other noble metals even further stimulates the search for inexpensive and easily available Ni-based catalysts. Although the field of nickel catalysis has rapidly expanded over the last decade, there are many challenges that remain to be overcome. Nickel catalysts retain significant synthetic potential, are very reactive and design/control of their catalytic systems requires much more effort. Indeed, in the majority of known Ni-mediated reactions, the active catalyst remains unknown. We expect to see further developments in the area of Ni-catalyzed C‒P bond formation, particularly in the expansion of substrate scope and the development of low-cost, air-stable and easy-to-handle sources of nickel for catalysis.

## Figures and Tables

**Figure 1 molecules-26-05283-f001:**
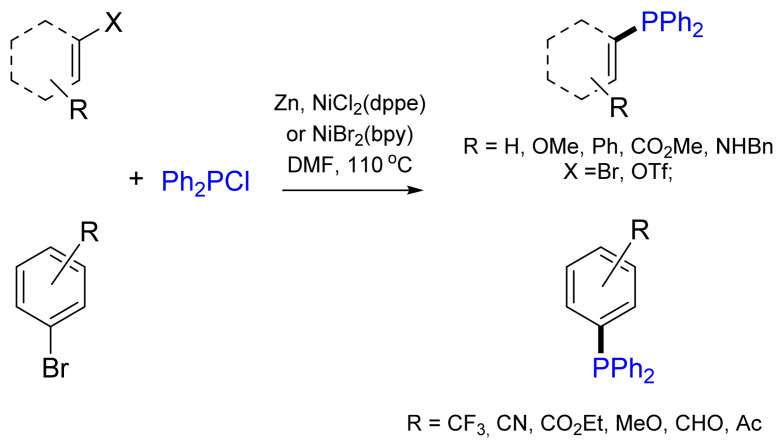
Nickel-catalyzed cross-coupling in the presence of zinc.

**Figure 2 molecules-26-05283-f002:**
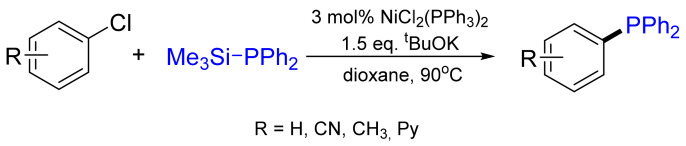
Nickel-catalyzed cross-coupling with (trimethylsilyl)diphenylphosphane.

**Figure 3 molecules-26-05283-f003:**
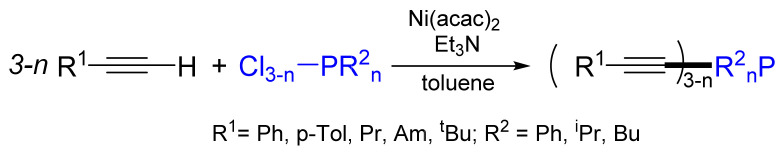
General scheme for a nickel-catalyzed cross-coupling reaction with terminal alkynes.

**Figure 4 molecules-26-05283-f004:**
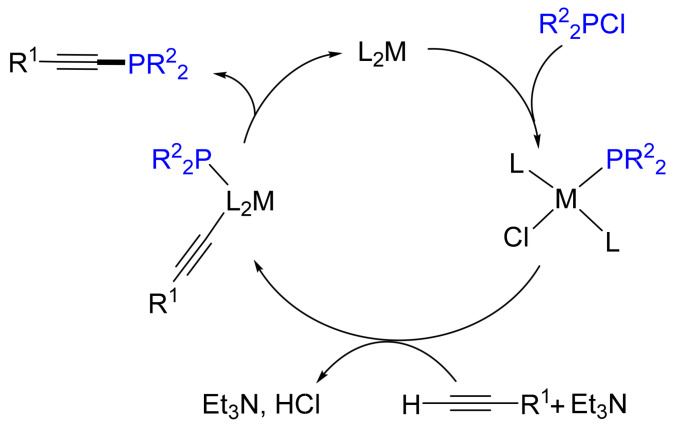
Proposed mechanism for the nickel- and palladium-catalyzed C(sp)-P cross-coupling (M = Ni, Pd).

**Figure 5 molecules-26-05283-f005:**
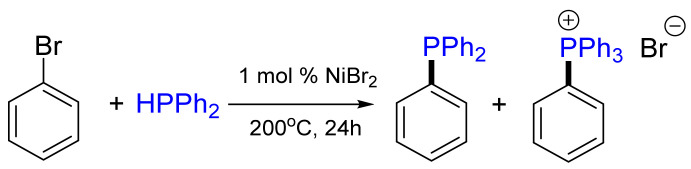
Nickel-catalyzed arylation of diphenylphosphane.

**Figure 6 molecules-26-05283-f006:**
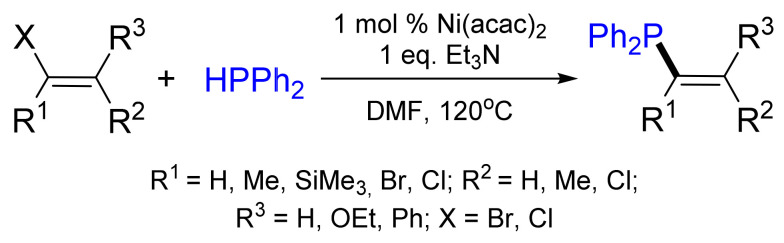
Nickel-catalyzed phosphination of vinyl bromides.

**Figure 7 molecules-26-05283-f007:**
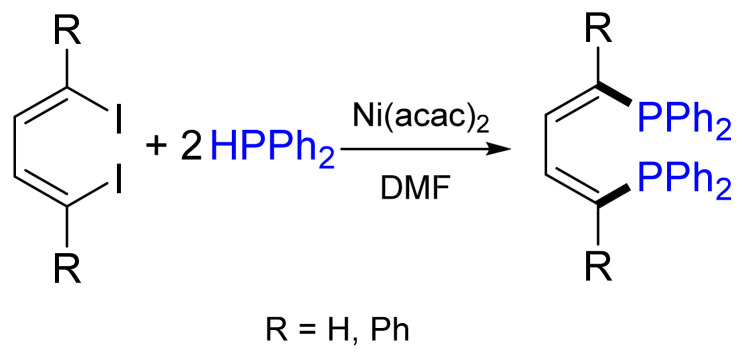
Nickel-catalyzed phosphination of vinyl iodides.

**Figure 8 molecules-26-05283-f008:**
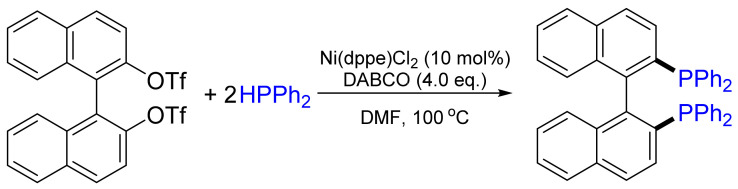
Ni(dppe)Cl_2_-catalyzed BINAP synthesis.

**Figure 9 molecules-26-05283-f009:**
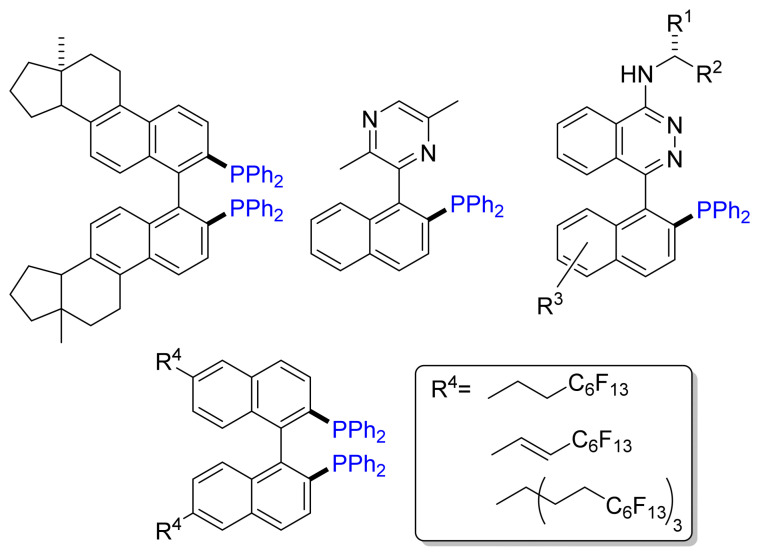
Selected examples of phosphane ligands obtained by Cai’s method.

**Figure 10 molecules-26-05283-f010:**
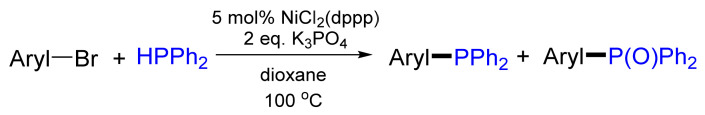
Nickel-catalyzed cross-coupling between aryl bromides and diphenylphosphane.

**Figure 11 molecules-26-05283-f011:**
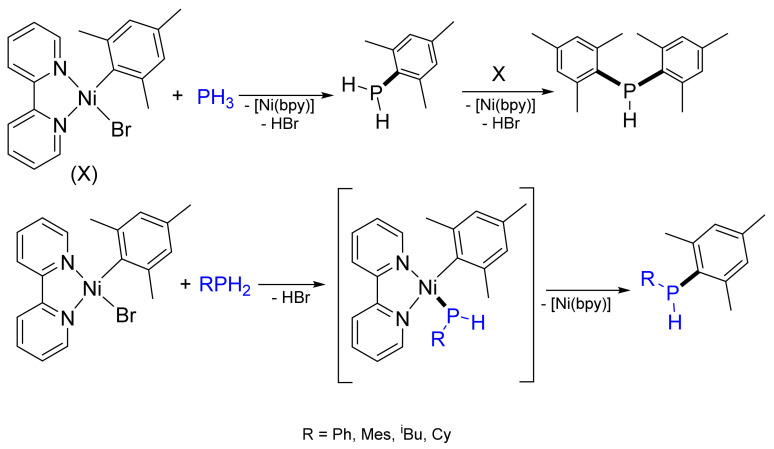
Reactivity of organonickel sigma-complexes with phosphanes.

**Figure 12 molecules-26-05283-f012:**
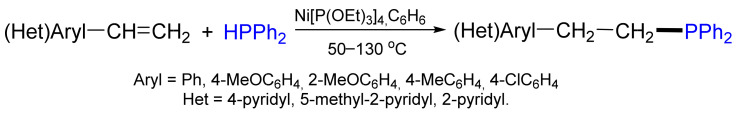
Hydrophosphination of weakly activated olefins and their heterocycle-containing analogues.

**Figure 13 molecules-26-05283-f013:**
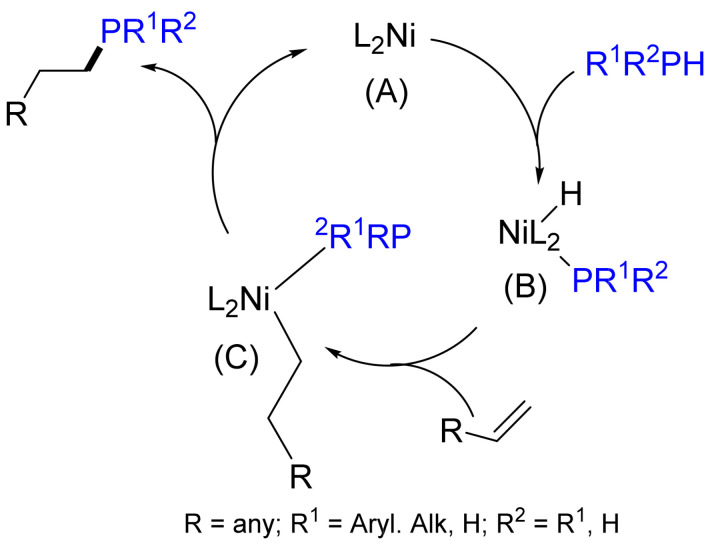
Mechanism of a Ni-catalyzed hydrophosphination of alkenes.

**Figure 14 molecules-26-05283-f014:**
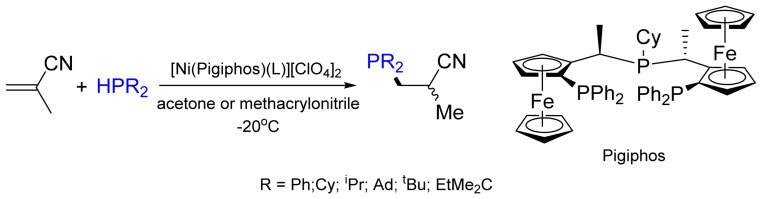
Ni-catalyzed asymmetric hydrophosphination of methacrylonitrile.

**Figure 15 molecules-26-05283-f015:**
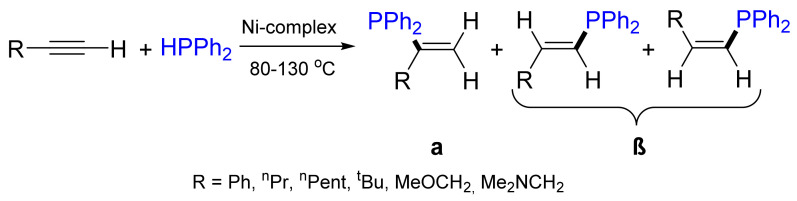
Nickel-catalyzed addition of HPPh_2_ to alkynes.

**Figure 16 molecules-26-05283-f016:**
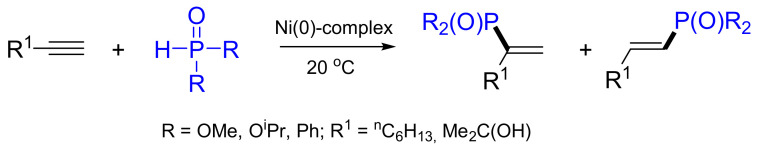
Nickel-catalyzed addition of R_2_P(O)H to alkynes.

**Figure 17 molecules-26-05283-f017:**
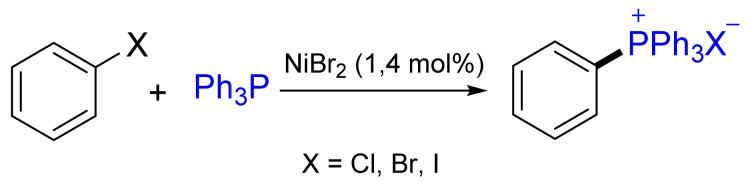
Nickel-catalyzed quaternization of triphenylphosphane.

**Figure 18 molecules-26-05283-f018:**
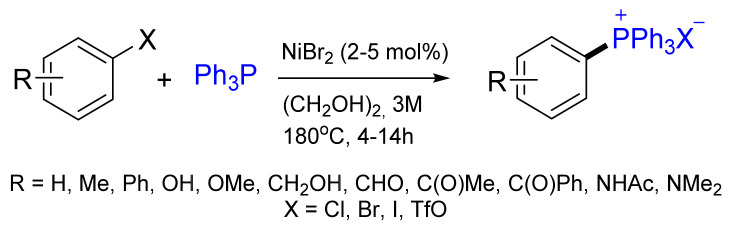
Scope of the Ni-catalyzed C‒P coupling.

**Figure 19 molecules-26-05283-f019:**
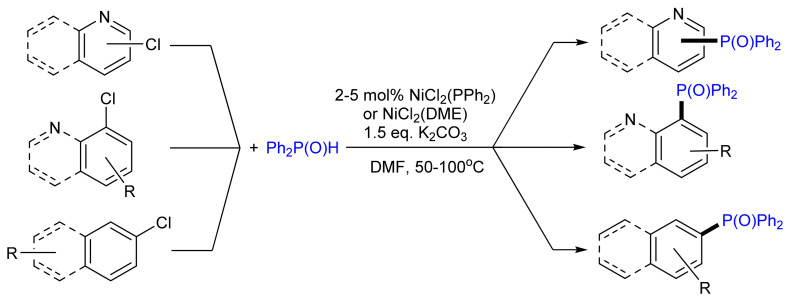
Nickel-catalyzed coupling of aryl chlorides with a diphenyl phosphite.

**Figure 20 molecules-26-05283-f020:**
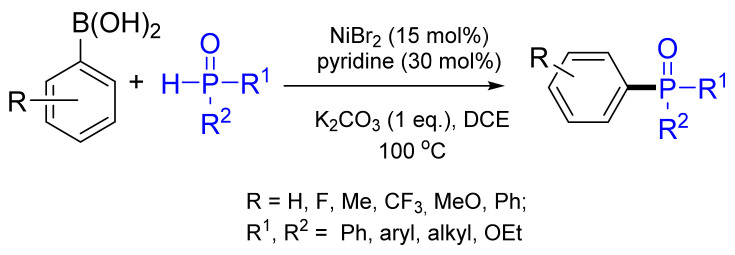
Nickel-catalyzed coupling of arylboronic acids with *P*-nucleophiles.

**Figure 21 molecules-26-05283-f021:**
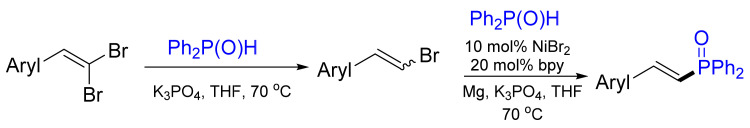
Reaction of alkenyl bromides with diphenylphosphane oxide.

**Figure 22 molecules-26-05283-f022:**
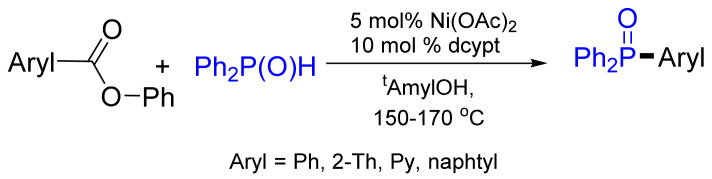
Decarbonylative coupling of esters with phosphane oxide.

**Figure 23 molecules-26-05283-f023:**
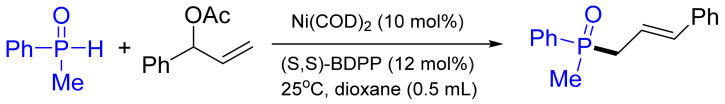
Ni-catalyzed asymmetric allylation of secondary phosphane oxides.

**Figure 24 molecules-26-05283-f024:**
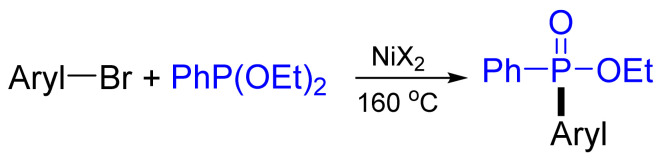
Ni-catalyzed reaction of phosphonic ethers with aryl bromides.

**Figure 25 molecules-26-05283-f025:**
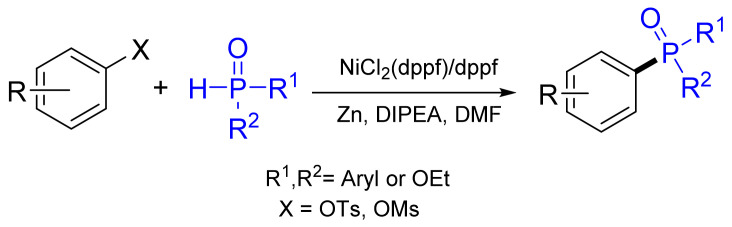
C‒P cross-coupling using aryl tosylates or mesylates.

**Figure 26 molecules-26-05283-f026:**
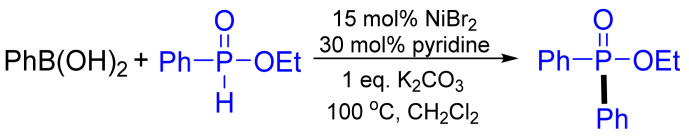
Reaction of phenylboronic acid with ethyl phenylphosphinate.

**Figure 27 molecules-26-05283-f027:**
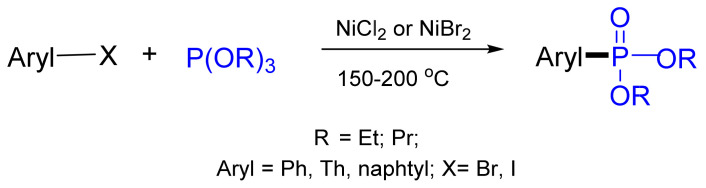
Cross-coupling reactions of aryl halides with triethyl phosphites and diethyl phenylphosphonites.

**Figure 28 molecules-26-05283-f028:**
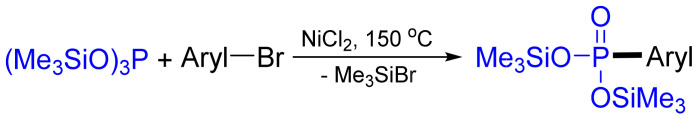
The catalytic arylation of tris(trimethylsilyl)phosphite.

**Figure 29 molecules-26-05283-f029:**
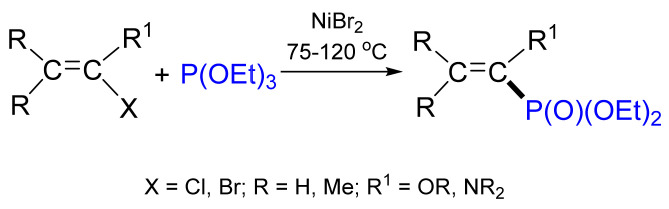
The reactions of triethylphosphite with alkenyl halides having an alkoxy or diethylamino group.

**Figure 30 molecules-26-05283-f030:**
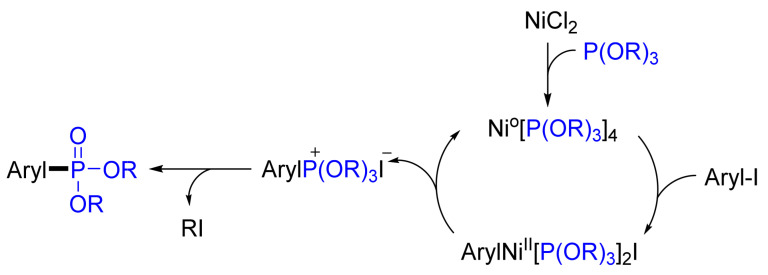
Proposed mechanism for the Ni-catalyzed cross-coupling of trialkyl phosphites with aryl iodides.

**Figure 31 molecules-26-05283-f031:**
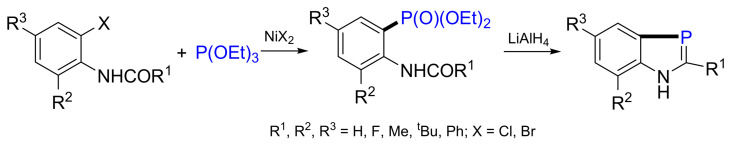
Ni-catalyzed reaction of triethylphosphite with 2-haloanilides.

**Figure 32 molecules-26-05283-f032:**
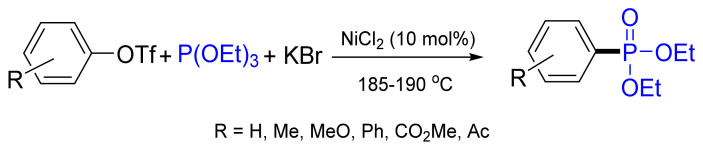
Nickel-catalyzed C‒P cross-coupling of aryl triflates with triethyl phosphite.

**Figure 33 molecules-26-05283-f033:**
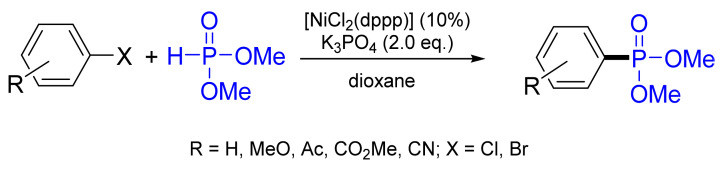
Cross-coupling of dimethyl phosphite with various aryl halides.

**Figure 34 molecules-26-05283-f034:**
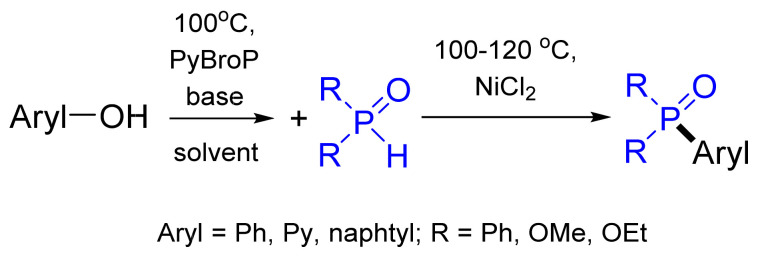
Hirao reaction after activating the hydroxyl group of phenols.

**Figure 35 molecules-26-05283-f035:**
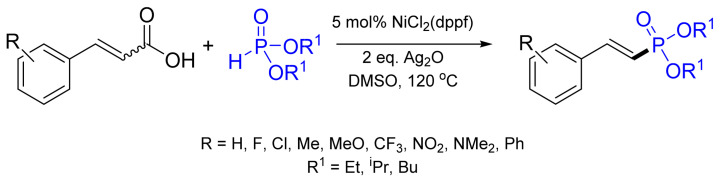
Decarboxylative cross-coupling of alkenyl acids with P(O)H compounds.

**Figure 36 molecules-26-05283-f036:**
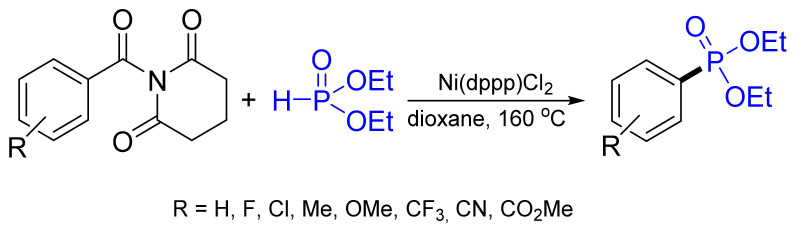
Phosphorylation of amides using Ni catalyst.

**Figure 37 molecules-26-05283-f037:**
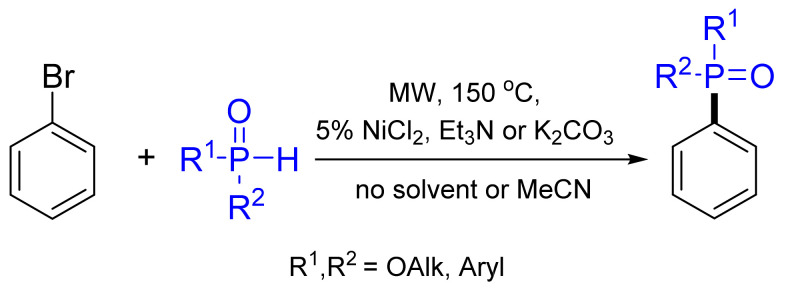
NiCl_2_-catalyzed phosphonylation.

**Figure 38 molecules-26-05283-f038:**
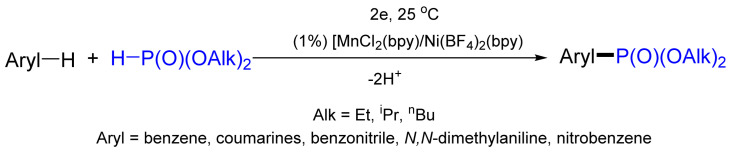
Phosphorylation of aromatic compounds in electrochemical conditions.

## Data Availability

Not applicable.

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
