# Peer review of "Nickel Complexes in C‒P Bond Formation"

_molecules, 2021, doi:10.3390/molecules26175283_

Round 1
Reviewer 1 Report
In this review, Yakhvarov and coworkers summarized the recent research progress in the formation of C‒P bonds derived by the cross-coupling reactions with the participation of nickel catalysts. This report covers most of the contemporary work done in this field, and some classic examples of related nickel-catalyzed reactions produced phosphanes, phosphonium salts, phosphane oxides, and phosphorus acid derivatives have been systematically (to some extent) outlined. However, the literature in this field is not fully cited (25 pages, 108 citations), and some obvious works have been missed (Nickel-catalyzed addition of H‒P(O) bonds to alkynes: 10.1021/ja0494297, 10.1021/ol0508431; Nickel-catalyzed synthesis of phosphonium salts: 10.1002/adsc.200800542, 10.1039/c6cc08938k; and etc. The author should carefully investigate relevant research in the field). In spite of this, I still believe that this review has the potential to be suitable for publication in molecules but not acceptable for possible publication in its current form.
The following major issues should be addressed by authors before the acceptance:
- The classification of the secondary titles in this article is inappropriate and the citation of related literature is not comprehensive: The research content of the full-text review is divided into two categories (chapters 1 and 2), among which section 1.2 introduces the arylation reactions, but the contents of Figures 6 and 7 appear here. The content of section 2.1 is too simple and should be enriched. In addition, the mechanism of nickel-catalyst participation in the reaction and the role of Ni in it should be discussed, although the author has summarized and prospected in the article. Overall, it is recommended to reorganize the research work in recent years and allocate it reasonably.
- The table of contents should be added before the introduction, so that the reader can understand it at a glance.
- In Figure 1, "NiBr2(bpy)" should be added to the reaction conditions, the two types of products do not use the same Ni-catalyst.
- In Figure 2, tBuOK → tBuOK
- The correct use of (which type of) hyphens must be double-checked, e.g., the hyphen for P‒C bonding should use "‒" instead of "-".
- In the caption of Figure 7, the incomplete statement should be perfected.
- In Figure 12, the labeling of the solvent (toluene) in the reaction should be consistent with the literature (benzene). Please check the cited literature carefully.
- In Figure 15, please carefully check the literature to confirm the type of Ni-catalyst. In addition, the type of product (α, β) should be consistent with the text (page 8, line 211-213).
- On page 13, line 325, "73%" should be indicated as yield.
- In Figure 33, please carefully check the literature to confirm the molar ratio of Ni-catalyst.
- In the reference section the references need to be carefully checked and consistent in writing format, especially the text before and after "doi", please write "https://doi.org/" uniformly.
Reviewer 2 Report
In this manuscript D.G. Yakhvarov and coworkers carried out a comprehensive update on the use of nickel complexes in C-P bond formation. This review details the recent advances in the synthesis of a variety of phosphorus products, revising some aspects on the proposed catalytic cycles involving nickel complexes as catalytic precursor. In general, the manuscript is well organized, and the thematic is suitable for the readers interested on the catalytic synthetic approaches for the obtainment of phosphorus compounds. Therefore, it can be accepted for publication in Molecules after minor changes.
- It might be better to understand if the reference numbers were also cited in all figures.
- It is desirable to include in each figure, the number of examples and yields range, for avoiding subjective terms as high yields.
- Page 7, line 177: …efficiency, what makes it… -> efficiency, that makes it…
- Page 8, figure 14: must be acetone and methacrylonitrile
- Page 11, line 291: … Ni(OAc)2 and was demonstrated -> Ni(OAc)2 was demonstrated
- Page 11, line 293: the yield was high (48-82%). I consider that the yields are modest to good.
- Page 12: Revise the caption of figure 12
- Page 16, figure 32: Consider including the activated Aryl-OH intermediate in the reaction scheme.
Round 2
Reviewer 1 Report
The authors have revised the paper accordingly, the new version is now acceptable.